# Factors Affecting Attitudes towards COVID-19 Vaccination: An Online Survey in Slovenia

**DOI:** 10.3390/vaccines9030247

**Published:** 2021-03-12

**Authors:** Luka Petravić, Rok Arh, Tina Gabrovec, Lucija Jazbec, Nika Rupčić, Nina Starešinič, Lea Zorman, Ajda Pretnar, Andrej Srakar, Matjaž Zwitter, Ana Slavec

**Affiliations:** 1Faculty of Medicine, University of Maribor, 2000 Maribor, Slovenia; rok.arh@student.um.si (R.A.); tina.gabrovec@student.um.si (T.G.); lucija.jazbec@student.um.si (L.J.); nika.rupcic@student.um.si (N.R.); nina.staresinic@student.um.si (N.S.); lea.zorman@student.um.si (L.Z.); matjaz.zwitter@guest.arnes.si (M.Z.); 2Faculty of Computer and Information Science, University of Ljubljana, 1000 Ljubljana, Slovenia; Ajda.Pretnar@fri.uni-lj.si; 3Institute for Economic Research, 1000 Ljubljana, Slovenia; andrej.srakar@ier.si; 4InnoRenew CoE, Livade 6, 6310 Izola, Slovenia; ana.slavec@innorenew.eu

**Keywords:** cross-sectional studies, intention, public opinion, trust, ordinal regression, COVID-19, vaccination, surveys and questionnaires, Europe, immune system, SARS-CoV-2

## Abstract

While the problem of vaccine hesitancy is not new, it has become more pronounced with the new COVID-19 vaccines and represents an obstacle to resolving the crisis. Even people who would usually trust vaccines and experts now prefer to wait for more information. A cross-sectional online survey was conducted in Slovenia in December 2020 to find out the attitudes of the population regarding COVID-19 vaccination and the factors that affect these attitudes. Based on 12,042 fully completed questionnaires, we find that higher intention to get vaccinated is associated with men, older respondents, physicians and medical students, respondents who got the influenza vaccination, those who knew someone who had gotten hospitalised or died from COVID-19 and those who have more trust in experts, institutions and vaccines. Nurses and technicians were less likely to get vaccinated. In answers to an open question, sceptics were split into those doubting the quality due to the rapid development of the vaccine and those that reported personal experiences with side effects of prior vaccinations. Although the Slovenian population is diverse in its attitudes towards vaccination, the results are comparable to those found in other countries. However, there are potential limitations to the generalizability of the findings that should be addressed in future studies.

## 1. Introduction

While vaccination is widely recognised as an effective way to reduce or eliminate the burden of infectious diseases by health authorities and the medical community [1], its benefit also depends on the willingness of individuals to be vaccinated [2]. This is the case with the COVID-19 vaccines that might be key to stopping the COVID-19 epidemic.

Vaccine hesitancy, i.e., the unwillingness of a proportion of the population to get vaccinated, is closely related to public trust in the health system [3]. To develop an effective communication strategy to advance vaccine readiness, it is crucial to understand the various socio-demographic factors that affect the decision [4]. It is particularly important to find out the positions of those on the vaccination priority list such as different groups of healthcare professionals (HCP).

An online survey about the attitudes towards COVID-19 vaccination in Slovenia was conducted in December 2020, just before the arrival of the first doses of the vaccine. In addition to finding out how many would take the vaccine, the study aimed to estimate the factors that impact the decision. These results can provide healthcare policymakers in Slovenia and other countries reliable information to allow them to improve immunisation plans and create better-targeted vaccination-related communication campaigns.

## 2. Methods

A cross-sectional study was designed to study public opinion towards COVID-19 vaccination in Slovenia. The target population of the survey were Slovenian residents older than 15. To assure high standards of data quality, the questionnaire was pretested on 17 people before being fielded. Test subjects used the comment function to note unclear wordings, enabling us to improve questions and answers to more understandable and straight-forward. In total, 12 comments were made and were addressed.

The questionnaire was developed based on previous studies [2,5,6,7] and had 11 closed-ended questions regarding the opinion on vaccination in general, attitudes towards different news sources and personal perception of lockdown orders as well as one open-ended question to entail everything a person might want to add. At the end, participants were asked seven demographic questions (gender, age group, education level, work status, profession, region and type of dwelling). Professions were classified as either non-healthcare professionals (non-HCP) or the healthcare professionals (HCP) identified as physicians, other HCP (nurses and technicians), medical students and HC students (health care students). An English translation of the questionnaire is added in Appendix A. 

The survey was conducted with the OneClick Survey (v 20.12.03, RRID:SCR_019283) tool and different media outlets were used to disseminate the survey link. To improve the representativeness of the nonprobability online sample, a wide array of media and other organisations (*n* = 167) were contacted and asked to disseminate the survey link. The survey invitation is added as Appendix A.

Analysis of quantitative data was done with SPSS (release 27.0.0.0, RRID:SCR_019096), except for ordinal regression and mediation analysis that was done in STATA (release 15.0, RRID:SCR_012763). The sample was described using frequency distributions and bivariate cross-tabulations and correlations were computed for pairs of variables. To reduce the dimensionality of data for scale questions, a principal component analysis (PCA) was carried out to compute dimensions to be used in further analysis. Based on the literature review, we developed a theoretical regression model with the intention to get vaccinated as the dependent variable that we were able to operationalise with our survey data. We ran ordinal regression to estimate the effect of a set of independent variables with the intention to get vaccinated; the model included gender, age, education, previous vaccination against influenza, HCP and PCA components.

In addition, we performed an analysis of textual data using Orange Data Mining (release 3.27.1, RRID:SCR_019811). The answers for a subsample of the respondents who completed the free-form answer were preprocessed with lowercase transform, tokenization by words, lemmatization with UDPipe 2 lemmatizer [8], stop word removal and removal of digits. TF-IDF transform was used for vectorization of documents. Clustering was performed with cosine distance and hierarchical clustering with Ward linkage. Large subgroups were further explored with another layer of clustering with the same parameters. We analysed the clusters with Chi2, Student’s t-test and ANOVA for determining the difference between groups (either a selected subgroup versus all or subgroups between one another). To determine the content of each group’s answers, we used a mixture of close and distant reading. 

An in-depth description of the methods according to the CHERRIES checklist for reporting results of online surveys [9] can be found in Appendix A.

## 3. Results

### 3.1. Survey Response

Of the 169 contacted organisations, at least 20 have shared the survey link (possibly more but they did not confirm by e-mail). We estimate the sum potential reach of these organisations is more than 400,000 users (assuming a zero overlap in followers) which is about 48% of all social media users according to the data of the Statistical office of Slovenia. However, organic posts usually do not reach all users and according to some estimates the average reach is only between 5–6% [10] of the followers, i.e., between 20,000 and 24,000.

However, in our case the reach was definitely higher since there were 45,633 unique clicks on the survey link in the eleven days that the survey ran, December 17–27 in 2020. Of those who clicked, 27,356 did not start to respond after landing on the survey page, while 18,277 started to respond but 3765 dropped out and so only 12,042 completed the survey. We estimate the response rate is between 3% (assuming the survey link reached more than 400,000 people) and 26% (assuming the reach was the same as the number of clicks).

### 3.2. Demographic Characteristics

The majority of respondents were female (63%), had high education (64%) and most of them were relatively young (42% were below 35, 40% from 35 to below 54 and 18% were 55 or older). Since many among the contacted individuals or organisations were health-related, 2068 respondents (17%) were HCP. The demographic data of HCP in our sample specifically compared with national registry can be found in Appendix A. Demographic data are presented in Table 1.

### 3.3. Intention to Get Vaccinated 

Among all respondents, 3964 (33%) indicated that they definitively intended to participate in vaccination and an additional 3119 (26%) replied that they would probably agree to vaccination; in total, 59% intended to vaccinate. As shown in Figure 1, the intention was similar for non-HCP (57%) and some groups of HCP (51% HC students, 50% other HCP), while physicians (84%) and medical students (82%) were significantly more inclined towards vaccination (*p* < 0.001, c = 0.158).

More men than women definitively intended to participate in vaccination (66% vs. 55%, *p* < 0.001, Chi Square = 148.487 Contingency Coeff. = 0.110). Likewise, a positive attitude towards vaccination increased with age category (23% for ages 15–25 vs. 62% for 75+, *p* < 0.001, Chi Square = 324.499, Contingency Coeff. = 0.162) (Figure 2).

In the past, only 18% of respondents had been regularly vaccinated against influenza. A strong association was observed between prior vaccination against influenza and a positive attitude towards vaccination against SARS-CoV-2 (43% vs. 78%, *p* < 0.001, Chi Square = 1542.675, Contingency Coeff. = 0.337) (Figure 3).

Respondents who knew someone who was hospitalized or died due to the virus (44% of all respondents) were more likely to accept vaccination (66%) as compared those who did not know anyone with that experience (53%) (*p* < 0.001, Chi Square = 202.946, Contingency Coeff. = 0.129).

### 3.4. Principal Component Analysis 

The questionnaire included 11 questions about trusting information from different sources and seven items about agreement with different COVID-19 vaccine statements. We extracted three components for which the eigenvalue was above 1. The threshold was decided based on the change in slope observed in the scree plot of eigenvalues (i.e., after the third component it gets flatter). Together, these three components explain 60% of the variance.

To interpret the meaning of variables, we looked at their weights (Table 2). The first component had the highest positive weight (0.86) for trust in the National Institute of Public Health (NIPH) and the safety and efficiency of the vaccine, followed by trust in the Ministry of Health (0.79), the World Health Organisation (WHO) and opinion of experts (0.78), news on television and radio (0.77), professional articles and research results (0.74) and daily newspapers (0.72). At the same time, weights for the first component had a high negative value for believing that vaccination is an attempt to control the population (−0.79) and having negative experiences with prior vaccinations (−0.61). We call this component “trust in official sources”. Nevertheless, it should be noted that this component has non-negligible weights for some items that are not completely aligned with this name (i.e., being scared of getting infected).

The second component had a high positive value for trusting information from friends and acquaintances that are non-HCP (0.74) and alternative explanations on social media (0.66). To a lesser extent, it also involved trust in information from those friends and acquaintances that are HCP (0.53). We call this component “trust in alternative sources”.

The third component had a strong negative weight for trust in the government (−0.63) and, to some extent, a strong desire to wait for more information about the safety of the vaccine (0.47). We call this component “distrust in government”.

We computed mean component values for different HCP (Appendix A). Physicians and medical students have a positive mean value for trust in official sources and negative for trust in alternative sources, while other HCP, HC students and those respondents that are non-HCP have a negative mean value for trusting official sources and positive for alternative sources. The differences for the third component were less pronounced. Figure 4 shows the scatterplot of values for the first and second component.

### 3.5. Ordinal Regression

We ran an ordinal regression to estimate the relationships between the intent to vaccinate and gender, age, education, experience with COVID-19, profession and the three dimensions extracted with PCA. The results are presented in Table 3.

The regression coefficients are significant for age, gender, education, being a physician and the three PCA components. Respondents who were older, male, physicians and those who trust official sources were more ready to get vaccinated, while more educated respondents, those who trust alternative sources and distrust the government were more vaccine hesitant. It should be noted, however, that for education the result is in a different direction than expected, so we performed a sensitivity analysis that showed (unlike other variables) education is sensitive to different settings; thus, we interpret the evidence as inconclusive. Knowing someone who had been hospitalized or had died due to COVID-19, being other HCP, medical student or HC student did not have a significant effect on the intention to get vaccinated.

Although prior vaccinations were also strongly associated with COVID-19 vaccination intention, we did not include it in the above ordinal regression model because it has a strong relationship with all other variables in the model (Appendix A).

Moreover, mediation analysis was used to correct for the effect of prior influenza vaccinations, deriving both the direct effects of relationships as well as the indirect ones through their effect on prior vaccinations. The summary results are shown in Table 4. The mediation effect is the highest for physicians (100%), which means that being a physician only affects the intention to get vaccinated indirectly through prior influenza vaccinations. In other words, if you are a physician, you are more prone to get vaccinated whether it is for influenza or COVID-19.

The second highest effect, but much lower, is for education (33%), which means that only 67% of the negative effect from higher education on COVID-19 vaccination is direct, while one third is indirect, through influenza vaccination. Thus, the regression coefficient in Table 2 can be reduced by 33% to −0.135 (as shown in the last column of Table 4). The effect is considerable for age (18%) and gender (12%), while it is rather small (5–6%) for the three components.

In addition, since plots in Figure 4 indicate there is an interaction between the PCA components and healthcare profession, we conducted an additional analysis to moderate it. Results show that being a physician or other HCP increases the trust in official resources and lowers distrust in government, while for trust in alternative resources there are no significant effects.

### 3.6. Analysis of Textual Data

We analysed responses from the 2320 respondents (12%) who answered the open-ended question. Compared to others, they had lower trust in the WHO (2.8 average vs. 3.3, *p* < 0.01, Student’s *t*-test = 17.4), with a larger percentage of people opposing vaccinations (17.8% vs. 6%, ddof = 2, Chi square = 321.4). They had a lower trust of NIPH (3 average vs. 3.45, *p* < 0.01, Student’s *t*-test = 15.6) and a higher ratio of past bad vaccine experience (2.5 average vs. 2.1, *p* < 0.01, Student’s *t*-test = 13.6).

We used hierarchical clustering to discover groups of similar responses. After preprocessing, we were left with 7625 unique tokens (words), which we transformed into a TF-IDF weighted bag of words. We computed the distance matrix with cosine distance and used Ward linkage for clustering groups. While the dendrogram did not reveal a good number of clusters (Figure 5), we decided on 10 clusters and further explored larger ones.

The first three clusters are most different from the rest as they join the larger cluster at later stages. The larger bottom subcluster forms two groups, the first containing clusters C4 and C5 and the second containing the remaining clusters. We used word enrichment to determine which words best define each cluster (Appendix A).

After determining the initial ten clusters, we further explored the two largest clusters, namely C5 and C10. C5 mostly contains notes on vaccination, such as questions about vaccine safety for risk groups and worries about side effects (Table 5). 

Cluster C10 contains comments about government measures as well as a sizable number of sceptics (Table 6).

In summary, clusters C3 and C8 represent the sceptics. These respondents were also in higher agreement that the vaccine is an attempt to control the population, that the virus is the same as influenza and were against vaccinations in general. These two groups reported trusting alternative explanations on social media to a higher degree than the official sources (Q6.7). 

Cluster C6 represents pro-vaccine respondents, which also had a higher agreement that the vaccine is safe and efficient. Respondents in C6 and C1 plan to get vaccinated as soon as possible (Q9). The former cluster has higher trust in official sources, while the latter in experts.

To validate the findings and determine the context of significant words, we also read a couple of responses in each cluster. An interesting finding was that many sceptics used scientific language to convey their doubt in the vaccine. For example, a large number of people claimed that the virus had not yet been isolated according to the Koch postulate. The second group of sceptics pointed to the role of the human immune system and how it should have been the primary line of defence. The third group of sceptics listed their personal bad experiences with vaccines as the reason why they were against vaccination. Among the milder sceptics, there were many comments regarding length of immunity after the vaccine, responsibility for the side effects of the vaccine and safety of the vaccine for people with allergies.

Even the pro-vaccine respondents with academic degrees stressed the importance of educating the public and presenting the findings. For instance, one of the physicians wrote: “Medical science should take people with bad vaccine experience more seriously. They should design measures to explain why there was an adverse reaction to the vaccine, how it can be prevented further, and how the people who indeed have bad reactions are the ones who need the rest of the population to have a high vaccination rate.”

## 4. Discussion

In terms of sample size, this survey is the biggest national study on attitude towards vaccination carried out in any European state to date [4]. Among all respondents, 59% answered that they will definitely or probably get vaccinated, which is about the same as in the USA in November 2020 (60%) [11], lower than China (91%), Australia (85%) and most Western and Northern European countries [4,12,13,14] but higher than most Eastern and South-Eastern European countries [4]. However, it should be noted that most of the surveys had been conducted in spring 2020 and the attitudes have probably changed since then.

A positive attitude towards vaccination is more prevalent among men and increases with age, which was also confirmed in previous vaccination studies [2,12,15]. Conversely, we were not able to confirm the finding that a high education level is associated with the intention to get vaccinated [15,16]. There are no studies exploring the reasons for gender and age differences in SARS-CoV-2 vaccine hesitancy, but we assume it can be explained by the fact that COVID-19 is more deadly for men and the elderly. However, there is some research for other diseases. For instance, a pneumococcal vaccine survey has shown that older people have more trust in vaccine and have more positive attitude towards vaccination based on former good experiences [17]. 

Studies in the USA and China have shown a strong association between prior vaccination against influenza and the intention to get vaccinated against SARS-CoV-2 [11,12], a finding which was confirmed by our study as well. An added value of our analysis was to include influenza vaccinations as a mediator variable, while the effects of other variables are lower than it appears on first sight. Another US study indicated that those who personally knew someone who had been hospitalized or had died as a result of having COVID-19 had a higher intention to get vaccinated compared to those without that experience [11], which was also confirmed by our crosstabulation analysis. A study about hypothetical Ebola vaccine had the same results as the USA study regarding COVID-19, the authors of the study go a step further and suggest that it is possible that knowing someone who got sick and witnessing their treatment and disease progress scares the individuals enough to make them more amendable to vaccination [18,19]. However, the effect of this variable is not significant in the ordinal regression model, which means we cannot generalise this association.

In contrast with other studies, the population in our study included a significant proportion of HCP and, thus, allowed us to analyse their attitudes towards vaccination. The attitudes and behaviour of HCP’s related to COVID-19 vaccination are of great importance as people are more likely to get vaccinated if recommended by their HCP [20]. In France, their intention to get vaccinated was higher among HCP than non-HCP [21]; while a survey in the USA showed that being employed as HCP was negatively associated with the intent to get a vaccine as compared with those who were never employed in HCP [20]. A survey in Los Angeles found that with nurses and personnel, regardless of patient contact roles, there are higher odds for delay or refusal of a coronavirus vaccine [22].

Discrepancies in findings of previous studies are probably due to the heterogeneity of HCP. Our study found that physicians and medical students have a higher trust in official sources and are more likely to believe in the safety and efficiency of the vaccine, while other HCP and HC students are more likely to believe alternative sources, even more than non-HCP. Nevertheless, the results of ordinal regression show that, with the exception of physicians (indirectly), HCP does not have an effect on vaccine readiness.

There are many reasons explored by other authors on vaccine hesitation among HCP. A review in 2015 found that HCP vaccination rates vary between different infectious agents and between different specialities [23]. A qualitative study on nurses found that they prefer conventional health beliefs rather than evidence-based medicine [24]. This is in line with our finding that many of the other HCP trust alternative sources mor than official ones [24].

A general trust in science and experts, together with accepting that the vaccine as safe and efficient, is the predictor of intention to get vaccinated against COVID-19 [25,26], while alternative explanations for the pandemic, distrust in science and government, as well as an unstable political situation, play a major role in vaccine rejection [4,22,27]. An American survey has indicated that the trust not only in government but also in science is significantly higher among those who endorse vaccination [6]. Similarly, a European survey found that the leading reasons for people choosing not to get vaccinated are not believing in the efficacy of vaccine, believing to have influenza is better than vaccination, having influenza-like illness despite vaccination, believing that influenza is not fatal and experiencing vaccine-related adverse events [28]. Moreover, a UK survey found that low confidence in the health system to handle the pandemic was associated with greater mistrust of vaccine safety and more worries about the unforeseen vaccine effects, whilst low confidence in government to handle the pandemic was associated with lower scores on worries about unforeseen effects and preference for natural immunity [29]. Furthermore, a Polish survey points out that lack of trust in public authorities could increase support towards conspiracy theories and lead to insufficient vaccination rate [30].

In our study, we included several items about trust and other attitudes from these previous studies and confirmed that they are affecting vaccination intention. With PCA, we extracted three dimensions: trust in official sources, trust in alternative sources and distrust in government. The intention to get vaccinated is positively associated with the first one and negatively with the other two. 

The analysis of open-ended responses provided us with a deeper insight into how respondents think about vaccinations and offers practical suggestions on how to communicate with sceptics. For instance, some of them like to use scientific language to present their statements, despite incorrect or irrelevant theories. Another group of sceptics likes to point out the role of the human immune system and yet another brings up personal bad experiences with vaccines such as side effects. Both were also mentioned in other studies [29,31,32]. Thus, it is important to take them seriously and address these issues when communicating with the public.

The study has some potential limitations. Although the sample was much larger than in any of the previous national studies, it should be pointed out that we used non-probability sampling (as in most of the previous research, with a few exceptions) so the findings should not be generalized to the whole population. Nevertheless, we were able to validate almost all presented findings on the probability panel sample that used the same questionnaire on a sample of 512 respondents and was proven to be representative of the Slovenian population [33], but did not include a large enough number of HCP to be used for the main study.

Having a cross-sectional study design, we are limited to a specific time frame and are not able to estimate how attitudes change in time. Another drawback is being limited to only one country in Europe; however, previous studies did not show any significant differences between countries, so we can assume that the findings in a national context have wider application.

Furthermore, our questionnaire did not include all the relevant factors mentioned in previous studies. For instance, we could have also included questions about socioeconomic status [6,11,15], awareness of herd immunity [34] and fear of side effects [31,32]. The latter were highlighted in our survey’s open-ended response results.

## 5. Conclusions

Our study confirmed findings in previous literature that respondents who intend to get vaccinated are most likely to be male and older. Those who knew people that were severely affected by the illness are likewise more inclined to take a shot. The intention is also positively associated with more trust in official sources of information, such as experts and public health institutions, while trust in alternative sources and mistrust of government makes for a lower chance that a person will agree to vaccination.

Being the largest public opinion survey in a single country to date, we were also able to gather the data for HCP and have demonstrated that physicians and medical students are more in favour of vaccination when compared to other HCP.

The above-mentioned effects can be broadly explained by prior influenza vaccinations that act as a mediating variable. This is a finding specific to our study as previous research has not applied mediation analysis to their models.

Moreover, open-ended question responses revealed that vaccine hesitancy is related to the rapid vaccine development and scepticism about its efficiency. The main focus in future efforts to vaccinate the global population should address all of the above-mentioned factors.

Further research should include questions about the fear of side effects and other relevant aspects that were revealed to be important for understanding the attitudes of the COVID-19 vaccination sceptics.

## Figures and Tables

**Figure 1 vaccines-09-00247-f001:**
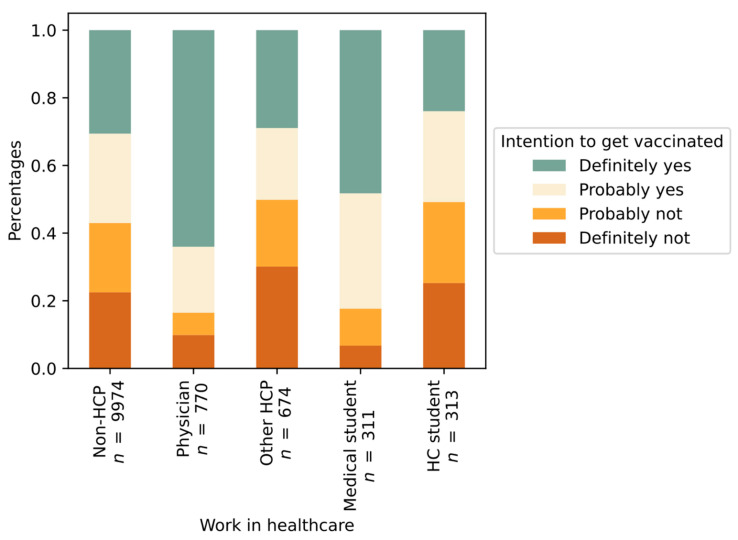
Intention to get vaccinated against COVID-19 grouped by working status in healthcare.

**Figure 2 vaccines-09-00247-f002:**
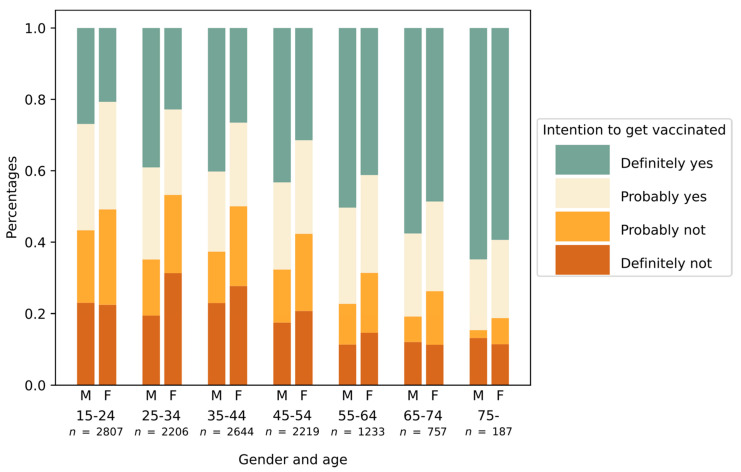
Intention to get vaccinated by age group and gender (M: male, F: female).

**Figure 3 vaccines-09-00247-f003:**
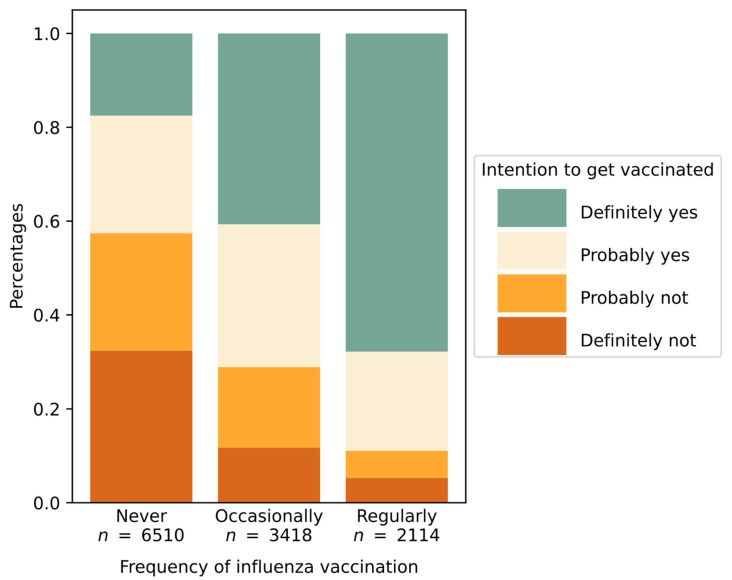
Intention to get vaccinated grouped by frequency of past influenza vaccination with shown subdivision regarding the COVID-19 vaccination intention.

**Figure 4 vaccines-09-00247-f004:**
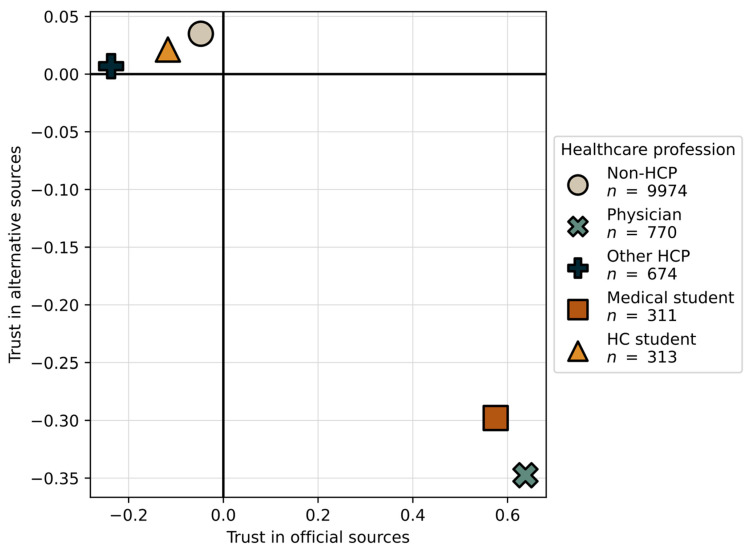
Positioning of healthcare profession in scatterplot of first and second PCA component.

**Figure 5 vaccines-09-00247-f005:**
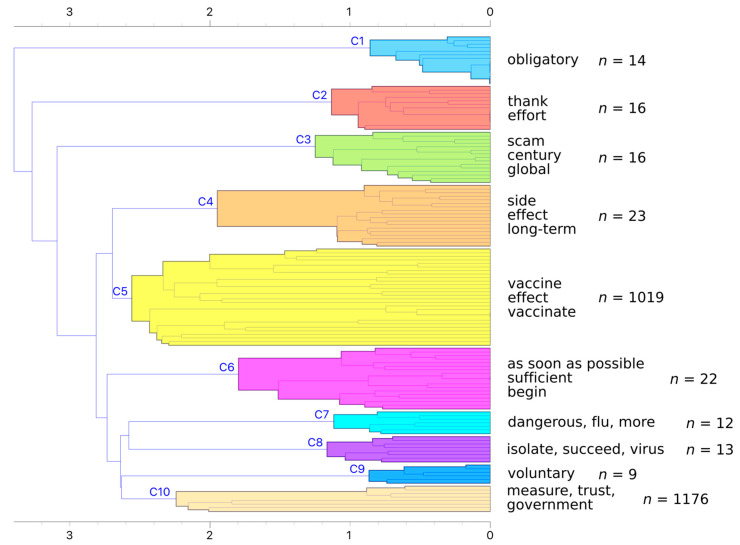
Dendrogram of hierarchical clustering with Ward linkage. The dendrogram is pruned at depth 10 for compact view.

**Table 1 vaccines-09-00247-t001:** Demographic data; percentages calculated within the column for gender and age separately (* nurses, caregivers, physiotherapists, pharmacists, psychologists, etc.; ** students of nursing faculties or high schools, other medicine-related fields of study).

Category		Non-HCP	Physicians	Other HCP *	Medical Students	HC Students **	Total
Gender	Female	6026	537	568	230	266	7627
		60.4%	69.7%	84.3%	74.0%	85.0%	63.3%
	Male	3948	233	106	81	47	4415
		39.6%	30.3%	15.7%	26.0%	15.0%	36.7%
Age (yr)	15–24	2192	7	47	274	287	2807
		22.0%	0.9%	7.0%	88.1%	91.7%	23.3%
	25–34	1686	242	223	35	20	2206
		16.9%	31.4%	33.1%	11.3%	6.4%	18.3%
	35–44	2264	175	192	1	1	2633
		22.7%	22.7%	28.5%	0.3%	0.3%	21.9%
	45–54	1940	156	119	1	3	2219
		19.5%	20.3%	17.7%	0.3%	1.0%	18.4%
	55–64	1035	119	79	0	0	1233
		10.4%	15.5%	11.7%	0.0%	0.0%	10.2%
	65–74	679	66	12	0	0	757
		6.8%	8.6%	1.8%	0.0%	0.0%	6.3%
	≥75	178	5	2	0	2	187
		1.8%	0.6%	0.3%	0.0%	0.6%	1.6%
Total		9974	770	674	311	313	12,042
		100.0%	100.0%	100.0%	100.0%	100.0%	100.0%

**Table 2 vaccines-09-00247-t002:** Principal component analysis, a matrix of weights.

Principal Component Analysis (PCA): Matrix of Weights		Component		
		1	2	3
*Trust in sources of information*	Q6a Reports on television and radio.	0.771	0.202	0.192
	Q6b Daily newspaper.	0.718	0.220	0.259
	Q6c National Institute of Public Health.	0.859	0.115	−0.133
	Q6d The ministry of health of the Republic of Slovenia.	0.793	0.196	−0.302
	Q6e World Health Organization (WHO).	0.785	0.090	0.152
	Q6f The government of the Republic of Slovenia.	0.556	0.186	−0.626
	Q6g Alternative explanations on social media.	−0.366	0.662	−0.183
	Q6h Professional articles and research findings.	0.741	0.082	0.285
	Q6i Expert opinion.	0.776	0.134	0.208
	Q6j Information provided to me by acquaintances employed in the field of healthcare.	0.396	0.527	0.131
	Q6k Information from friends and acquaintances that are not employed in the field of healthcare.	−0.070	0.738	−0.073
Agreement with statements	Q10a I trust that the vaccine against SARS-CoV−2 virus is safe.	0.858	−0.160	−0.079
	Q10b I believe that vaccination against SARS-CoV-2 virus is effective.	0.851	−0.145	−0.016
	Q10c I would like to wait for more information on the safety of the vaccine against SARS-CoV-2 virus.	−0.214	0.292	0.473
	Q10d I am very scared of getting infected with SARS-CoV-2 virus.	0.471	0.024	−0.209
	Q10e I think that SARS-CoV-2 virus is equally dangerous as the influenza virus.	−0.583	0.262	0.055
	Q10f I have negative experiences with vaccinations—considering me or my loved ones.	−0.608	0.255	−0.078
	Q10g Vaccination against SARS-CoV-2 virus is an attempt of controlling the population.	−0.786	0.257	−0.023

**Table 3 vaccines-09-00247-t003:** Results of ordinal logistic regression for intention to get vaccinated (NA—not applicable).

Intention to Get Vaccinated against COVID-19	Coef.	Std. Err.	Z	*p* > |z|	[95% Conf. Interval]	
Age (1 = 15–24, 2 = 25–34,…, 6 = 65–74, 7 = 75+)	0.487	0.028	17.170	0.000	0.431	0.542
Gender = male	0.283	0.041	6.960	0.000	0.203	0.363
Education = high	−0.200	0.043	−4.680	0.000	−0.283	−0.116
Experience with COVID-19 (i.e., knew someone who got hospitalised or died from it)	0.015	0.039	0.380	0.701	−0.062	0.092
Profession = physician	0.301	0.089	3.390	0.001	0.127	0.476
Profession = other HCP	0.123	0.084	1.460	0.144	−0.042	0.289
Profession = medical student	0.205	0.122	1.680	0.093	−0.034	0.443
Profession = HC student	0.136	0.118	1.150	0.249	−0.095	0.368
Component 1: Trust in official sources	2.464	0.032	77.280	0.000	2.402	2.527
Component 2: Trust in alternative sources	−0.582	0.021	−28.250	0000	−0.622	−0.541
Component 3: Distrust in government	−0.203	0.020	−10.040	0.000	−0.243	−0.163
/cut1 |	−1.699	0.064	NA	NA	−1.825	−1.573
/cut2 |	0.318	0.061	NA	NA	0.199	0.438
/cut3 |	2.469	0.065	NA	NA	2.341	2.597

**Table 4 vaccines-09-00247-t004:** Results of mediation analysis.

Intention to Get Vaccinated against COVID-19	% of Total Effect Mediated	[95% Conf. Interval]		Regression Coef. Calculated Based on Direct Effect
Age (1 = 15–24, 2 = 25–34,…, 6 = 65–74, 7 = 75+)	11.80%	10.55%	13.28%	0.395
Gender = male	18.19%	13.88%	26.00%	0.231
Education = high	32.72%	20.65%	77.91%	−0.135
Profession = physicians	100.00%	82.83%	355.30%	0.000
Component 1: Trust in official sources	5.63%	5.54%	5.71%	2.325
Component 2: Trust in alternative sources	4.81%	4.51%	5.16%	−0.554
Component 3: Distrust in government	5.13%	4.26%	6.49%	−0.193

Note: The last column is the value of the coefficient in Table 1 reduced for the mediated effect.

**Table 5 vaccines-09-00247-t005:** Detailed analysis of cluster C5. It contains two subclusters, those that are eager to get the vaccine and those that have second thoughts about it.

Subcluster	Description		No. of Answers
C5.1	“When will we be able to get the vaccine?”		91
	C5.1.1	Vaccine and risk groups	37
	C5.1.2	Vaccine availability	54
C5.2	Second thoughts		928
	C5.2.1	“I don’t know.”	7
	C5.2.2	“We are not test rats.”	37
	C5.2.3	Methodological reservations about the survey	128
	C5.2.4	“I am worried about side effects.”	756

**Table 6 vaccines-09-00247-t006:** Detailed analysis of cluster C10. It contains roughly five subclusters, but it is fairly diverse. It contains both respondents that wish to get vaccinated quickly (C10.3) and those that believe the vaccine is a Big Pharma scam (C10.5).

Subcluster	Description	No. of Answers
C10.1	“Don’t get vaccinated.”	7
C10.2	“Experts should be in higher agreement.”	172
C10.3	“We need a vaccine as soon as possible.”	15
C10.4	“Vaccine is a genocide.”	11
C10.5	Conspiracy theories	917

## Data Availability

The data presented in this study are available via the Slovenian Social Science Data Archives repository (https://www.adp.fdv.uni-lj.si/eng/, accessed on 28 January 2021) at https://doi.org/10.17898/ADP_SARSPR20_V1 and https://doi.org/10.17898/ADP_SARSVE20_V1 [33,35].

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
