# Peer review of "Factors Affecting Attitudes towards COVID-19 Vaccination: An Online Survey in Slovenia"

_vaccines, 2021, doi:10.3390/vaccines9030247_

Round 1

Reviewer 1 Report

I think this is an interesting and important paper, which will be relevant to the readers of Vaccines. I have some minor comments and suggestions that, in my opinion, may improve some aspects of the paper.

  • What’s the survey response rate? You provide absolute numbers of people answering the survey, but what is the denominator, i.e the group exposed to the survey? This might be hard/impossible to even estimate. However, low response rates can be indicative of bias, as you have rightfully discussed.
  • Another potential way of asserting that your survey is not highly biased (although it is more of a way to find out if it is biased) is to compare the demographic characteristics (e.g. age and gender) of HCP categories to their actual distribution (e.g. whether 74% of the entirety of medical students are indeed female, or is it very far off). This data should be easily available from the Bureau of Statistics equivalent in Slovenia. I think this will strengthen the study.
  • The PCA explanation is a nice touch! However, PCA component number 1 had other, non-negligible weights (e.g. I am very scared of getting infected…), that are not completely aligned with the “name” you gave it. Same goes for other components. I would mention that the PCA is not perfectly aligned with your interpretation.
  • From what you saw in the PCA plots, I would guess that the interactions between some of the variables and the HCP of the respondents would have an interesting effect.
  • The supplementary has tracked changes left in…
  • A switch in fonts occurred in the middle of the abstract
  • Abstract – “…higher intention is…” intention of what?
  • Introduction – “the COVID-19vaccines” add space (this repeats several times)
  • “Vaccine hesitancy, i.e., the unwillingness of a substantial proportion…” – it could also refer to a small proportion, the definition doesn’t have to do with frequency
  • “from the age of 15” – older than
  • “The questionnaire was developed based on previous studies…” - elaborate
  • “The questionnaire was pretested on 17 people before being fielded.” – what were the results or the meaning of this pretesting? What was gained/changed?
  • “…intention to vaccinate as the dependent variable…” – do you mean the intention to get vaccinated?
  • You use a period “.” to signify thousands, the common way in English is a comma “,” (e.g. ten thousand=10,000)
  • “…there were 45.633 unique clicks on the survey link. In total, 18.277 started to respond and 12.042 completed the survey.” – that doesn’t add up, what happened with the rest? They clicked and didn’t even start to respond? Please elaborate.
  • “(p>0.001, c=0.158)” - I guess you meant <0.001, otherwise it might not be statistically significant…
  • Also, when providing a p-value, please state the test used
  • Why was the threshold of an eigenvalue of 1 set? Were the data normalized before the PCA?
  • “…had not yet been isolated according to the Koch postulate…” – do you mean has not been determined as the cause of the disease?
  • “However, the effect of this variable is not significant in the ordinal regression model, which means we cannot assume a causal relationship.” –even if it were, deduction of causality from such association is very very risky. I would rephrase. This pertains to other places where causality-oriented language is used instead of association measures.
  • “At variance with other surveys” – ?

Author Response

Response to Reviewer 1 Comments

Point 1: I think this is an interesting and important paper, which will be relevant to the readers of Vaccines. I have some minor comments and suggestions that, in my opinion, may improve some aspects of the paper.

Response 1: We would like to kindly thank the reviewer 1 for her/his suggestions. We hope that our revisions are appropriate.

Point 2: What’s the survey response rate? You provide absolute numbers of people answering the survey, but what is the denominator, i.e the group exposed to the survey? This might be hard/impossible to even estimate. However, low response rates can be indicative of bias, as you have rightfully discussed. 

Response 2: According to the data of the Statistical Office of Slovenia, the internet is regularly used by 83 % of the population aged from 15 to 74 and 52 % are users of social media. The Slovenian population in this age category is about 1,590,426, we estimate there are about 1,320,426 internet users in Slovenia and about 827,0022 social media users. This is the population that we are able to cover with an online survey.

We contacted 167 different organisations and asked them to share the link to the survey. This included newspapers, radio and television networks, both national and local, hospitals, municipalities, and some other organisations, including the website 24ur.com that is the most visited website in Slovenia (with more than 300,000 unique daily visitors). They shared the link on their Facebook site that has 234,571 followers which is 28 % of all the social media users in Slovenia. In total, at least 19 organisatons shared the link (possibly more but they did not confirm to us) and if we sum up the number of Facebook followers of these 19 organizations' (assuming there is no overlap), we get to the number 403,289 followers which is 49 % of social media users in Slovenia.

On one hand, the number should be higher as (at least in some cases) the link was not shared only on the Facebook page but also on other social media and/or by e-mail (but we don’t know with how many people). On the other hand, not all followers have been exposed to the survey but it is hard to estimate how many as the percentages of reach are different for different page sizes. According to some estimates the reach is only between 5-6 %; however, that is much too low in our case as it calculates to between 20,164 and 24,197 folowers and there were 45,633 unique click on the survey link. That is 19 % of the followers of the largest Facebook page that shared the survey and 11 % of the sum of followers for all 19 organisations (assuming the overlap is 0).

Of those that clicked the survey link, 12,042 (26 %) completed the questionnaire till the last page. If we consider the reach that might be anything between 45,633 and 403,289 people, the response rate is between 3 % and 26 %. However, neither assumption is realistic, they are just theoretical possiblities and the true value is most probably somewhere in the middle.

LOCATION: line 103

Point 3: Another potential way of asserting that your survey is not highly biased (although it is more of a way to find out if it is biased) is to compare the demographic characteristics (e.g. age and gender) of HCP categories to their actual distribution (e.g. whether 74% of the entirety of medical students are indeed female, or is it very far off). This data should be easily available from the Bureau of Statistics equivalent in Slovenia. I think this will strengthen the study. 

Response 3: We have contacted the Bureau of Statistics in Slovenia and aggregated demographic data of our country. The table with this data is published as supplement 4.

LOCATION: supplement 4

Point 4: The PCA explanation is a nice touch! However, PCA component number 1 had other, non-negligible weights (e.g. I am very scared of getting infected…), that are not completely aligned with the “name” you gave it. The same goes for other components. I would mention that the PCA is not perfectly aligned with your interpretation. 

Response 4: After naming the component we added a note that it has non-negligible weights for some other items that are not completely aligned with this name.

LOCATION: line 169

Point 5: From what you saw in the PCA plots, I would guess that the interactions between some of the variables and the HCP of the respondents would have an interesting effect. 

Response 5: We conducted an additional analysis to control for the interaction between the health profession and PCA components. Being a physician or other HCP increases the trust in official sources and decreases trust in government. We included this at the end of the ordinal regression section.

LOCATION: line 219

Point 6: The supplementary has tracked changes left in…

Response 6: We have accepted changes and checked all other supplements for this issue, none other was found.

LOCATION: supplementary

Point 7: A switch in fonts occurred in the middle of the abstract.

Response 7: This has been resolved.

LOCATION: line 21

Point 8: Abstract – “…higher intention is…” intention of what?

Response 8: We have added “intention to get vaccinated”, which now reads as “higher intention to get vaccinated”.

LOCATION: line 26

Point 9: Introduction – “the COVID-19vaccines” add space (this repeats several times)

Response 9: This was fixed throughout the document.

LOCATION: throughout the document

Point 10: “Vaccine hesitancy, i.e., the unwillingness of a substantial proportion…” – it could also refer to a small proportion, the definition doesn’t have to do with frequency

Response 10: This was fixed with the removal of the word substantial, it now reads as “Vaccine hesitancy, i.e., the unwillingness of a proportion of the population”.

LOCATION: line 44

Point 11: “from the age of 15” – older than 

Response 11: We have changed this to older than.

LOCATION: line 59

Point 12: “The questionnaire was developed based on previous studies…” - elaborate

Response 12: Previous studies in this field used in the preparation of the questionnaire were added as sources.

LOCATION: line 63

Point 13:  “The questionnaire was pretested on 17 people before being fielded.” – what were the results or the meaning of this pretesting? What was gained/changed?

Response 13: This is now briefly covered in the sentences after.

LOCATION: line 60

Point 14:   “…intention to vaccinate as the dependent variable…” – do you mean the intention to get vaccinated?

Response 14: Yes, thank you! This has now been resolved.

LOCATION: line 85

Point 15:  You use a period “.” to signify thousands, the common way in English is a comma “,” (e.g. ten thousand=10,000) 

Response 15: This has been resolved throughout the document.

LOCATION: throughout the document

Point 16:  “…there were 45.633 unique clicks on the survey link. In total, 18.277 started to respond and 12.042 completed the survey.” – that doesn’t add up, what happened with the rest? They clicked and didn’t even start to respond? Please elaborate.

Response 16: We have added and explained the difference between the numbers.

LOCATION: line 111

Point 17:  “(p>0.001, c=0.158)” - I guess you meant <0.001, otherwise it might not be statistically significant…

Response 17: This was fixed.

LOCATION: line 134

Point 18:  Also, when providing a p-value, please state the test used.

Response 18: Statistical tests were added.

LOCATION: throughout the document

Point 19:  Why was the threshold of an eigenvalue of 1 set? Were the data normalized before the PCA?

Response 19: We decided not only based on the eigenvalue but also based on the change in slope observed in the screen diagram. We added the explanation in the text.

LOCATION: line 157

Point 20:  “…had not yet been isolated according to the Koch postulate…” – do you mean has not been determined as the cause of the disease?

Response 20: This was not changed as it is a direct line from our responses, illustrating what the main responses were.

LOCATION: line 265

Point 21:  “However, the effect of this variable is not significant in the ordinal regression model, which means we cannot assume a causal relationship.” –even if it were, deduction of causality from such association is very very risky. I would rephrase. This pertains to other places where the causality-oriented language is used instead of association measures. 

Response 21: We changed the wording so that it says we cannot generalise this association and there is no mention of causality.

LOCATION: line 308

Point 22: “At variance with other surveys” – ? 

Response 22: This was changed to “In contrast with other studies ”.

LOCATION: line 310

Reviewer 2 Report

Dear authors,

Thank you for taking the initiative to conduct this study. Attitudes towards vaccinations have been a long-standing issue, but this has been brought to the forefront because of the COVID-19 pandemic and it's toll on lives globally.

There are many reasons why some people are anti-vaccine, and interestingly, this includes religious grounds. This study appropriately explores some of the reasons behind the study population's attitudes towards vaccination.

I commend the authors for contributing this manuscript, which should increase the understanding of public health professionals and policymakers on vaccinations and in crafting public messages.

Whilst the survey explored 'personal perception of lockdown orders', it wasn't clear to me how this impacted people's choice to vaccinate or not. Perhaps this could be made clearer.

It's befuddling why a higher percentage of HCP's do not have the intention to vaccinate. Some of the reasons are explored in this study but this could be explained further in the discussions.

Why do more older people and males have positive attitudes towards vaccination? This should be expanded upon in the discussion.

Why are "respondents who knew someone who was hospitalized or died due to the virus (44% of all respondents)..more likely to accept vaccination?" The ordinal regression found no significant effect of this experience on the intention to vaccinate. However, you could offer some explanation in the discussion. For example, is this due to fear or a better understanding of the importance of vaccines?

Last but not least, I appreciate the fact that this study does not call for blatant dismissal of those with misgivings about vaccines but suggests that these people should be taken seriously and their concerns addressed.

Author Response

Response to Reviewer 2 Comments

Point 1: "Thank you for taking the initiative to conduct this study. Attitudes towards vaccinations have been a long-standing issue, but this has been brought to the forefront because of the COVID-19 pandemic and its toll on lives globally. There are many reasons why some people are anti-vaccine, and interestingly, this includes religious grounds. This study appropriately explores some of the reasons behind the study population's attitudes towards vaccination. I commend the authors for contributing to this manuscript, which should increase the understanding of public health professionals and policymakers on vaccinations and in crafting public messages. "

Response 1: We would like to thank reviewer 2 for their kind words of encouragement and their acknowledgement of the importance of this topic.

Point 2: Whilst the survey explored 'personal perception of lockdown orders', it wasn't clear to me how this impacted people's choice to vaccinate or not. Perhaps this could be made clearer.

Response 2: We included this set of questions in the questionnaire on request from one of the stakeholders and since measures against the pandemic are very country-specific and there were no previous studies on this topic, we decided not to include it in the regression model. Nevertheless, we computed measures of association between the perceptions and the intention to vaccinate and found that those respondents who are the most bothered by the measures against the pandemic are less likely to get vaccinated. However, this is probably a spurious correlation and can be explained by the level of trust in official sources.
LOCATION: NA

Point 3: It's befuddling why a higher percentage of HCP's do not have the intention to vaccinate. Some of the reasons are explored in this study but this could be explained further in the discussions.

Response 3: HCP vaccine hesitation is an important public health problem. As per suggestion, we have extended this topic to include more peer-reviewed literature and explore the possible reasons behind it and how to address it.
LOCATION: line 325

Point 4: Why do more older people and males have positive attitudes towards vaccination? This should be expanded upon in the discussion.

Response 4: This is a very interesting question but we found no studies addressign the reasons for gender and age differences in COVID-19 vaccine hisitance. We assume it is due to the the disease is more deadly for men and the elderly. Moreover, we also added the finding of a study (for another disease) that the attitude of aloder poeple is based on former good experiences.
LOCATION: line 290

Point 5: Why are "respondents who knew someone who was hospitalized or died due to the virus (44% of all respondents)..more likely to accept vaccination?" The ordinal regression found no significant effect of this experience on the intention to vaccinate. However, you could offer some explanation in the discussion. For example, is this due to fear or a better understanding of the importance of vaccines?

Response 5: An interesting question, again the low number of peer-reviewed studies. We have added a paragraph addressing possible reasons that could explain why respondents that know the COVID-19 patients are more likely to accept the vaccine. A parallel was drawn with the EBOLA crisis in Africa.

LOCATION: line 303
